# The Etiological Diagnosis of Diabetes: Still a Challenge for the Clinician

**Danièle Dubois-Laforgue [1,2,3,*] and José Timsit [3]**

1   Department of Diabetology, Hôpital Cochin-Port-Royal, APHP, 123 Boulevard de Port-Royal, 75014 Paris, France
2   INSERM U1016, Université Paris Cité, 75006 Paris, France
3   PRISIS Reference Center for Rare Diseases, Université Paris Cité, 75006 Paris, France; jose.timsit@gmail.com
*   Correspondence: daniele.dubois@aphp.fr

**Abstract:** The etiological diagnosis of diabetes conveys many practical consequences for the care of patients, and often of their families. However, a wide heterogeneity in the phenotypes of all diabetes subtypes, including Type 1 diabetes, Type 2 diabetes, and monogenic diabetes, has been reported and contributes to frequent misdiagnoses. The recently revised WHO classification of diabetes mellitus includes two new classes, namely "hybrid forms" and "unclassified diabetes", which also reflect the difficulties of this etiological diagnosis. During the last years, many studies aiming at identifying homogenous subgroups on refined phenotypes have been reported. Ultimately, such subtyping may improve the diagnosis, prognosis, and treatment of patients on a pathophysiological basis. Here, we discuss the concepts of typical vs. atypical diabetes in the context of autoimmune Type 1 diabetes, Type 2 diabetes, and its monogenic forms. We discuss the contributions of clinical markers, biological tests, particularly islet cell auto-antibodies, and genetics to improving accurate diagnoses. These data support a systematic evaluation of all newly diagnosed diabetes cases.

**Keywords:** type 1 diabetes; type 2 diabetes; monogenic diabetes; MODY; phenotypic heterogeneity; endotype; etiological diagnosis; genetic diagnosis; next generation sequencing





## 1. Introduction

According to the World Health Organization, diabetes is a defined by the presence of chronic hyperglycemia, which can be considered as a symptom as well as a disease. Indeed, it has been long recognized that multiple etiologic and pathogenic pathways can lead to diabetes. As in other pathologic conditions (e.g., chronic fever) the etiological diagnosis of chronic hyperglycemia has important consequences for the care of patients, and for their families, and is a critical step in "precision medicine" or, more humbly, less imprecise medicine [1].

For a long time, the etiological diagnosis of diabetes was based solely on the clinical presentation of the patients upon their diabetes diagnosis (lean vs. obese, young vs. older), on a strict insulin dependency, or on the need, or not, for insulin therapy to treat the patient, the latter being obviously dependent on the availability of new and more efficient drugs to treat Type 2 diabetes. It has also been shown that most diabetes subtypes, including autoimmune Type 1 diabetes, Type 2 diabetes, pancreatic diabetes, and gestational diabetes, may progress from mild hyperglycemia to non-insulin dependent overt diabetes, then to a requirement for insulin therapy, depending on the progression of the beta cell disease and the deterioration of the residual insulin secretion [2]. Beyond these criteria, which remain widely used in clinical practice, technical advances, such as measurements of insulin secretion and insulin sensitivity, the identification of pancreatic beta cell autoimmunity, and genetic susceptibilities to several diabetes subtypes, have led to more refined classifications of diabetes.

These observations led to the WHO 1999 classification, which distinguished Type 1 diabetes, accounting for 5–10% of all diabetes cases, the vast majority being of autoimmune origin, Type 2 diabetes, accounting for 90% of cases, a series of "specific" forms, accounting for a minority of diabetes cases (e.g., pancreatic diabetes), and gestational diabetes. However, this classification still used both phenotypic and pathogenic criteria, did not account for the phenotypic variability of each diabetes subtype, and left some forms unclassified, such as ketosis-prone (Type 2) diabetes (KPD).

Following studies showing the heterogeneity of diabetes pathogeny and phenotypes (reviewed in [3]), the WHO classification was again revised in 2019 [4] and introduced two new categories:

- "Hybrid forms" of diabetes, including Slowly Evolving Immune-Mediated Diabetes (previously named Latent Autoimmune Diabetes in Adults, LADA) and Ketosis-Prone Type 2 Diabetes, the latter still being considered as a "non-autoimmune Type 1 diabetes" by the American Diabetes Association [5];
- "Unclassified Diabetes", i.e., cases with no ascribable definite etiology, particularly at the time of diagnosis.

Here, in this short, non-systematic review, we discuss the concept of typical/atypical forms of diabetes and highlight the difficulties in the etiological diagnosis of diabetes, which requires a systematic approach. These issues will be discussed in the context of Type 1, Type 2, and monogenic diabetes subtypes, as summarized in Figure 1.

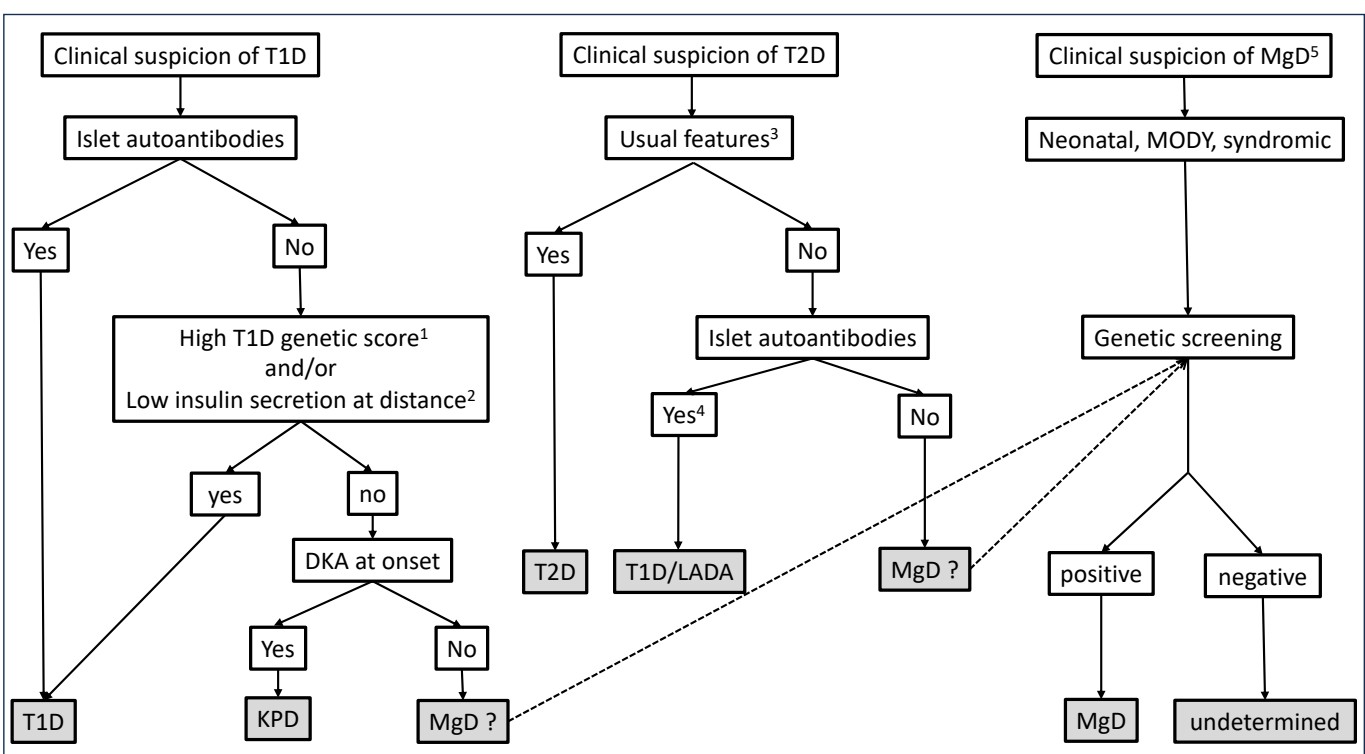

**Figure 1.** Suggested algorithms for etiological diagnosis of diabetes, based on clinical presentation at onset. [1] T1D genetic scores not yet available in routine practice. [2] Insulin secretion is informative when measured at a distance from diabetes onset and can be assessed by C-peptide plasma or urinary concentration. [3] Combination of criteria at onset/diagnosis of diabetes, such as age, body mass index, symptoms at diagnosis, non-insulin dependency at diagnosis, markers of metabolic syndrome, family history of T2D, history of gestational diabetes, and high-risk ethnicity. [4] Unless isolated low titers GADA. [5] In patients with no islet antibodies. Various phenotypes may suggest MgD, depending on genetic subtypes.

## 2. What Are the "Typical" Phenotypes of Type 1 and Type 2 Diabetes?

### 2.1. The Phenotype of Type 1 Diabetes Is Heterogenous

Type 1 autoimmune diabetes (T1D) is usually defined by a near-complete deficiency in insulin secretion due to the autoimmune destruction of beta cells, which is associated with the presence of anti-beta cell autoimmunity markers, particularly the autoantibodies directed to insulin, GAD, IA-2, and ZnT8. T1D occurs in the context of genetic susceptibility, mainly defined by certain HLA class II alleles [6]. Other genes or genetic markers have been shown to confer an increased T1D risk and have led to defining T1D genetic scores that may improve its diagnosis or prediction [7,8]. Insulitis is the pathologic hallmark of T1D and it is characterized by the presence of immune (mainly CD8+ T lymphocytes and B lymphocytes) and inflammatory cells within and around the pancreatic islets [9]. It has been shown to affect a small fraction of islets (10–30%), more often those containing insulin-positive beta cells, and to vary with age at the onset of diabetes and the disease duration. While autoantibodies are unlikely to play a direct role in beta cell destruction, their presence is highly predictive of the risk of diabetes in the relatives of T1D patients and in children from the general population, and has a good sensitivity and specificity for T1D [10]. It is now recommended that all adult patients with a phenotype that may suggest T1D should be tested for the presence of diabetes-associated antibodies, in order to confirm the diagnosis [11].

T1D is classically described as the rapid onset of the marked symptoms of hyperglycemia (polyuria) and weight loss in a lean child or young adult, or even a diabetic keto-acidosis (DKA) in 30–50% of cases, which is related to a profound deficiency in insulin secretion. This "typical" phenotype is probably restricted to a minority of patients with autoimmune T1D, and all of its characteristics may vary widely.

Indeed, although the incidence of T1D is increasing in very young children, the age of diagnosis is over 20 years in at least 50% of cases [12,13]. A study performed in a Euro-Caucasian population from Great Britain further showed that T1D, here defined by a high T1D genetic risk score [14], can occur at any age, with 40% of cases being revealed after the age of 30 years, accounting for 5% of all diabetes cases diagnosed after this age [15].

Such observations are not implemented in routine practice, as the same group reported that, among the patients with T1D occurring after 30 years of age, here identified by a marked loss in insulin secretion and the eventual need for insulin therapy, 40% had not received insulin from the onset of their diabetes and half of them thought they had T2D. The phenotypes of these patients at the onset of diabetes were, however, similar to those of patients in whom diabetes had occurred before 30 years, and 80% of them had anti-beta cell antibodies, even at a distance from diabetes onset [16]. Conversely, in another study, among 722 adult patients newly diagnosed with T1D on clinical grounds, 25% were antibody negative. Compared to those with antibodies, these patients had a lower T1D genetic risk score, a lower rate of loss of insulin secretion, and insulin could be stopped in 23% of them, with $HbA_{1c}$ values similar to those of antibody-positive patients [17]. Together, these studies show that differential diagnosis between T1D and T2D may be difficult in adults when based only on clinical features.

With regard to body weight, patients with T1D are not spared from the increased incidence of overweight and obesity observed in the general population. For example, in the USA, 30–35% of adolescents with T1D were overweight/obese at the time of their diabetes diagnosis and in the following years, a prevalence similar to or higher than that of the general population [18,19]. This may explain why young subjects with T1D may be misdiagnosed as having T2D and the increased frequency of diabetic ketoacidosis in these patients [20].

The degree of insulin deficiency at the diagnosis of T1D is also highly heterogenous, depending, for example, on the stage of the natural history or on the existence of regulatory mechanisms (inhibition) of the autoimmune anti-beta cell process itself. The clinical symptoms observed at the diagnosis of T1D may also vary according to the presence of some degree of insulin resistance. The "typical" revelation of T1D by DKA is probably

biased, too restricted, and a source of misdiagnoses. Indeed, diabetic ketoacidosis is not specific to T1D, which may lead to a spurious diagnosis of T1D. For example, DKA is, by definition, observed in Ketosis-Prone Diabetes, which is not associated with anti-beta cell autoimmunity [21]. The frequency of DKA at the onset of autoimmune T1D is highly variable. For metabolic reasons, it is higher in children (30–50%) and during puberty than in adults (10%), and higher in the presence of intercurrent infections. Moreover, risk factors for DKA independent of the pathogeny of T1D play a major role in its frequency. In a review of 24,000 T1D cases occurring in children, the frequency of DKA was increased by factors that affected care delivery: belonging to an ethnic minority, an absence of medical insurance, a low education level of the parents, misdiagnoses, and delayed diabetes treatment; two factors were protective: a family history of T1D and living in a region with a high prevalence of T1D [22]. In the same vein, studies in children at a high risk for T1D (relatives of patients with T1D or children at high genetic risk from the general population) have shown that the frequency of DKA at the onset of diabetes was reduced by a 5–10 factor when these children were regularly followed [23,24]. Thus, one frequent ("typical"?) phenotype of T1D could be the systematic discovery of mild hyperglycemia in a clinically asymptomatic individual, followed because of the presence of anti-beta cell autoimmunity on a high-risk genetic background [25].

### 2.2. The Case of LADA/Slowly Evolving Immune-Mediated Diabetes

At the other end of the clinical spectrum of T1D, slowly evolving forms have been described. In various populations, 5 to 10% of adult patients in whom the clinical presentation of diabetes at diagnosis suggests T2D, particularly in the absence of an insulin dependency, have anti-beta cell autoantibodies, mainly isolated anti-GAD antibodies (GADA) [3]. These observations have led to the description of Latent Autoimmune Diabetes in Adults (LADA), a diabetes subtype that is as prevalent as T1D and characterized by a progression to insulin requirement more rapid than that in T2D, suggesting a pathogenic role of autoimmunity. When patients are not systematically tested for the presence of autoantibodies and for residual insulin secretion, this diabetes subtype remains often misdiagnosed as T2D [26].

Similar observations have been made in children and adolescents. Two American studies (SEARCH and TODAY) reported the presence of GADA and/or IA-2 antibodies in 10 to 20% of children/adolescents aged 10–19 years in whom a diagnosis of T2D had been made on clinical grounds [27,28]. The phenotype of these patients was close to that of LADA, leading to the description of Latent Autoimmune Diabetes in the Young (LADY). As has been reported for adults, the clinical and metabolic phenotypes were different in the patients with and without antibodies, the phenotype of those with antibodies being closer to that of T1D. However, due to the overlap of these phenotypes, it was not possible, at the individual level, to distinguish the patients with autoimmune diabetes from those with T2D [28].

The genetics of LADA is characterized by an increased frequency of the susceptibility alleles associated with T1D (HLA class II) and of certain alleles associated with T2D [29,30]. A study showed that T1D-asssociated HLA class II genotypes, T2D-associated genotypes (in *TCF7L2* and *FTO*), and the presence of overweight/obesity synergistically interact to increase the risk of LADA [31].

Although these data led to the classification of this diabetes subtype as a hybrid form, namely "Slowly Evolving Immune-Mediated Diabetes", the existence of this subtype as a pathogenic entity remains controversial. For some authors, this diabetes subtype is considered as the pauci-symptomatic end of the pathogenic spectrum of T1D, while others favor the deleterious metabolic effects of insulin resistance in patients with a beta cell destruction less aggressive than that in T1D, or even arrested [3,32].

Indeed, there is some phenotypic heterogeneity among patients considered to have LADA: the majority have a normal body weight, while some are obese, although they have a lesser occurrence of metabolic syndrome than patients with classical T2D [26,33,34]. This led to a subclassification into LADA type 1 and LADA type 2, i.e., true autoimmune diabetes

with a T-lymphocyte-mediated destruction of beta cells in the former, and type 2 diabetes with "bystander auto-immunity", characterized by isolated GADA and no auto-immune destructive process in the latter [35]).

Data showing an increased loss in insulin secretion associated with the presence of GADA, as compared to that in patients without antibodies, are not always consistent [3,33]. In the same vein, the predictive value of isolated GADA is low, particularly when at low titers. In the relatives of T1D patients, those with isolated GADA have a low risk of progressing to diabetes at three years [36,37], and in subjects from the general population with no family history of diabetes, progression to diabetes at eight years was increased only in those with high GADA titers [38]. Additionally, isolated GADA have been shown to be transient in ~20% of children at high genetic risk for T1D [39], and false positive results have been suspected in many cases [35,40]. Moreover, histological studies have shown that asymptomatic subjects with isolated GADA have no insulitis [41] and no reduction in their beta cell mass [42]. Altogether, these data suggest that the diagnosis of LADA/Slowly Evolving Immune-Mediated Diabetes could be made by excess. GADA directed to the C-terminal epitopes of GAD65 or the N-terminal truncated GAD65 could provide a higher predictive value for GADA when isolated, but this assay is not yet affordable in common practice [43–45].

In total, many phenotypes may reveal T1D, including a severe DKA, the fortuitous discovery of moderate hyperglycemia in the context of an infection or a potentially diabetogenic treatment, the systematic screening of hyperglycemia in individuals at risk of T1D because of the existence of antibodies, a phenotype that may suggest T2D, or even gestational diabetes [46,47]. Thus, there is no unique "typical" phenotype of T1D.

The heterogeneity of T1D has led to consider whether the endotype concept, i.e., the heterogeneity of the disease phenotypes may result from differences in pathological mechanisms, can be applied to T1D. Many findings argue for this hypothesis (reviewed in [48]). For example, the degree of the immune infiltrate in the islets is more pronounced and contains more "hyper-immune" cells in early-onset diabetes, which has been shown to correlate with the degree of beta cell destruction and insulin secretion defects at diagnosis; the nature and number of auto-antibodies present at diagnosis also differ in children (multiple and predominantly IAA and IA2A) and adults (more frequently isolated and directed to GAD), and may be associated with different HLA haplotypes; also, the response to immune interventions may differ according to these traits, as recently suggested in a trial testing the protective effect of teplizumab in non-diabetic autoantibody-positive individuals [49]. Obviously, identifying T1D endotypes will have important consequences for the care of patients and their relatives, particularly with regard to the prediction of T1D and the immune interventions for preventing or altering the course of the disease.

### 2.3. T2D Subtyping: Which Consequences for the Clinician?

T2D is by far the most frequent form of diabetes, accounting for 90% of all diabetes cases, with its incidence increasing in all studied populations. The classical phenotype of T2D is described, in the absence of anti-beta cell autoimmunity markers, as a mild to moderate hyperglycemia, that is usually clinically asymptomatic in the early phase of the disease, with no insulin dependency, but worsens with time, occurring in middle-aged individuals, most often with a family history of T2D, or a maternal history of gestational diabetes. The risk factors for T2D include a non-Euro-Caucasian ancestry, an increasing age, a lack of physical activity, the presence of overweight/obesity, or an increased abdominal fat mass, and the presence of the insulin resistance markers of "metabolic syndrome", including a high blood pressure, high serum triglyceride levels, and low high-density lipoprotein levels. The occurrence and progression of T2D are due to qualitative and quantitative abnormalities of insulin secretion, which become insufficient to compensate for insulin resistances of variable severity [50,51]. Genetic and environmental factors are involved in the pathogeny of T2D, but the very first cause of the beta cell defect remains elusive.

All the characteristics of T2D, including its epidemiology, pathogeny, and phenotype, the prevalence of complications, and the response to treatment, are highly variable [52]. Moreover, there are no sensitive and specific markers that allow for a diagnosis of T2D, which remains a diagnosis by default.

During recent years, various strategies have been used to identify, among patients with a diagnosis of T2D, homogenous subgroups characterized by shared phenotypes, risks of complications, responses to available treatments, and presumably pathogeny ("endotypes") (reviewed in [53]).

In 2015, a study using a "topologic" analysis of many clinical and biological markers in T2D patients identified three clusters that differed in diabetes complications, associated co-morbidities, and the genetic variants that could account for it [54]. More recently, Swedish and Finnish investigators studied a population of 8980 adult patients with recent-onset diabetes. They used a cluster analysis based on six clinical and biological markers available at the time of diagnosis or at registration (age, BMI, $HbA_{1c}$, anti-GAD antibodies, insulin secretion, and sensitivity evaluated using HOMA2 methods) [55]. According to the classical criteria used for the classification of diabetes, two groups of patients would have been identified, namely T1D or LADA, defined by the presence of GADA and accounting for 6% of the total population, with the remaining patients considered as having T2D. Five clusters were identified, including "severe autoimmune diabetes" (SAID), defined by the presence of GADA and thought to include patients with T1D or LADA, and four non-autoimmune clusters, i.e., "severe-insulin deficient diabetes" (SIDD), "severe insulin-resistant diabetes" (SIRD), "mild obesity-related diabetes" (MOD), and "mild age-related diabetes" (MARD). These results were reproduced in three independent cohorts [55]. It is interesting to note that, except for the absence of GADA, the patients in the SIDD cluster had characteristics very similar to those of the patients in the SAID cluster, underlying the importance of antibody testing in this context [53]. Moreover, large overlaps of the clinical and biological criteria between the clusters and, within the same cluster, a large variability of each criterion were both observed (Figure 2).

Nevertheless, these analyses allowed for the identification of T2D subgroups with different phenotypes at diagnosis. Importantly, these subgroups could be associated with different pathogenic mechanisms (e.g., with regard to insulin secretion and insulin sensitivity), different risks of complications (e.g., retinopathy/neuropathy vs. nephropathy) [56], and different genetic markers [57]. Moreover, it has been recently reported that the levels of circulating metabolites or proteins may help to discriminate these clusters [58–60].

Many other studies have replicated these results in populations of various ancestries, although with differences in the proportions of the clusters in some, and in populations assessed at a distance from diabetes onset [53]. Whether these clusters could be universally identified in patients of various ethnicities should be confirmed, since different phenotypes, which could be associated with different pathogenic mechanisms, are suggested for some populations [61–63].

Since all these studies included patients with adult-onset T2D, it will be mandatory to perform studies in pediatric populations. Indeed, data from the USA indicate that, due to the increasing frequency of obesity and T2D in adolescents, in patients aged 15–19 years, the incidence of T2D is now higher than that of T1D in ethnic minorities (Afro-Americans, Hispanics, Indian Americans, and Hawaiians) [27,64]. Moreover, in these young patients, the progression from minor alterations in glucose tolerance to overt T2 diabetes is faster than that in adults [65].

Overall, these studies have confirmed the wide heterogeneity of T2D and the potential for clinical and biological markers available in routine, to identify subgroups among patients with T2D at diabetes diagnosis, which may respond to different pathological mechanisms and have different prognoses and responses to diabetes treatments. Thus, they exclude the existence of a single "typical" form of T2D and suggest that studies on patients with T2D, particularly intervention trials, should better take into account this heterogeneity, as mentioned in patients with T1D.

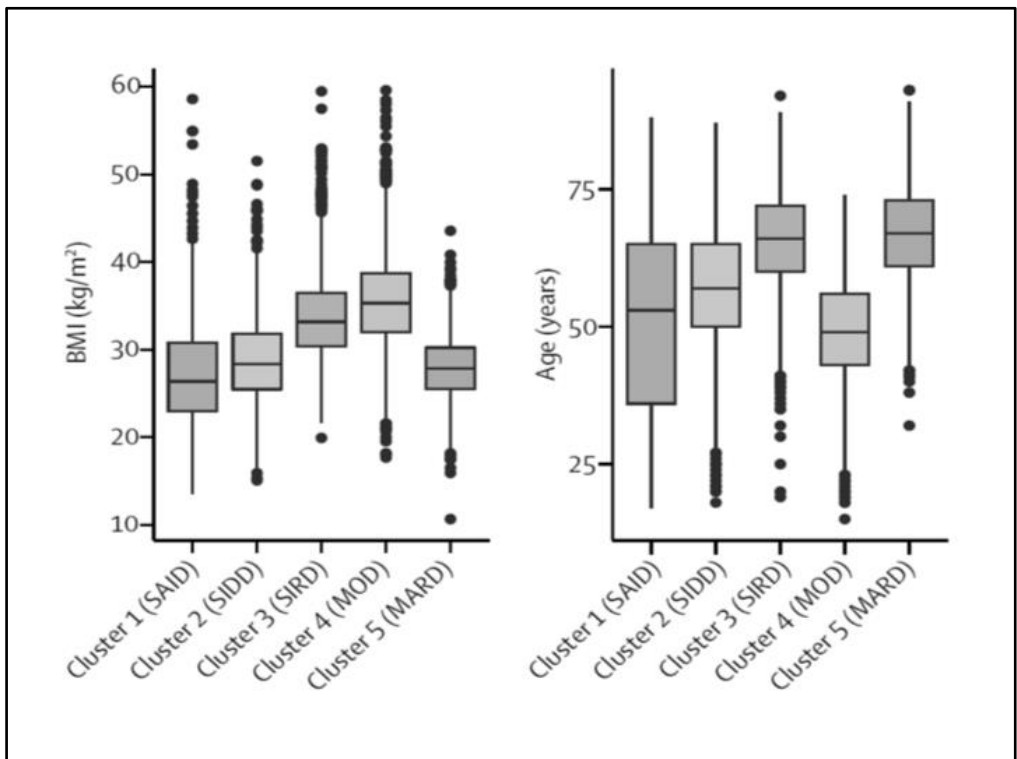

**Figure 2.** Body mass index and age at diagnosis in adult patients with diabetes. Using six clinical and biological criteria five clusters were identified among 8980 patients. The figure shows the distribution of body mass index (BMI) and age at diagnosis values in each cluster, and their overlaps between the clusters. Adapted from Ahlqvist et al. [55] with permission.

At the individual level, the clinical benefits of these subtyping studies for routine practice remain to be determined [1,53]. Particularly, it could be difficult to assign a given patient to one particular cluster. As an alternative, recent studies have shown that individual assessment based on the combination of clinical and biological variables can efficiently predict glycemic progression, the risk of chronic kidney disease, severe retinopathy and cardiovascular events, and the response to various treatments [66,67]. Additionally, a given patient may have characteristics belonging to several clusters. In a recent study, using a set of 32 clinical and biological parameters recorded in 726 patients with newly diagnosed T2D, four extreme phenotypes ("archetypes") were defined. They were associated with different genetics and omics profiles, different pathophysiological processes, and differences in disease progression (HbA$_{1c}$ values or a need for diabetes treatment). However, in the majority of the patients, a combination of at least two archetypes contributed to the phenotype [68]. Since the identification of these archetypes required in-depth phenotyping at the diagnosis of diabetes, which is not usually performed in a clinical setting, the relevance of these findings in routine care is not defined yet.

### 2.4. The Case of "Ketosis-Prone Diabetes"

Ketosis-prone diabetes (KPD) has been described in African American children who presented with DKA, but with no detectable anti-islet cell antibodies or HLA class II alleles associated with T1D, followed by remission of insulin dependency, in the context of weight excess and a strong family history of diabetes [69]. The same phenotype was reported under the term "Flatbush Diabetes" in patients of Sub-Saharan origin [70]. Thereafter, many studies have reported similar observations in patients with various ethnicities (reviewed in [71]).

The case of KPD, sometimes termed "atypical diabetes", illustrates well the limits of the etiological diagnosis of diabetes when it is based solely on clinical grounds. Indeed,

when patients presenting with DKA were more carefully phenotyped, by assessment of T1D antibodies, insulin secretion, and genetic markers, etiological heterogeneity was demonstrated. In a multiethnic study of 103 patients with DKA [72], four different subgroups were identified according to the presence or the absence of autoantibodies (A+ and A−) and of residual beta cell function (B+ and B−). The patients in the A+B− group (accounting for 21% of cases) had clinical and genetic features characteristic of T1D, including the absence of an improvement in beta cell function in the long term. Those in the A+B+ group (11%) had a phenotype suggesting T2D, but presented with DKA and were thought to have LADA. In this group, GADA were predominantly directed against the N-terminal epitope of GAD65, which was previously shown to correlate with preserved beta cell function [73]. The majority of the patients (50%) fitted into the A−B+ group, characterized, in this series, by a high frequency of Hispanic origin, a late onset (40 years on average), a high prevalence of overweight/obesity (70%), a family history of diabetes (88%), a low frequency of T1D at-risk HLA alleles, a high level of C-peptide at diagnosis that was even increased 6 months later, and a high rate of weaning off insulin at 6 months from the diagnosis, all of which were suggestive of T2D, leading further to the designation of "Ketosis-Prone Type 2 diabetes" (KPT2D). The A-B+ subgroup was further subdivided into two groups according to the presence, or not of a precipitating factor of DKA. The patients with "unprovoked" DKA may have a distinct syndrome with clinical and biological characteristic suggesting T2D, a better long-term beta cell function, and a long-term insulin independence with good glycemic control [74].

Whether KPD should be considered as a definite hybrid form of diabetes, sharing both T1D, but not autoimmunity, and T2D clinical characteristics, or only represent a severe presentation of T2D, is still under debate [21,75]. Several pathophysiological hypotheses have been investigated to determine whether KPD represents a discrete subtype of diabetes, but no firm conclusion has been attained [75]. Some authors have suggested that, in these patients, ketosis could be the result of a decrease in ketolysis, rather than an increase in ketones production [76].

### 3. Monogenic Diabetes: A Multi-Faceted Diabetes Subtype

Monogenic diabetes (MgD) has been first described under the term Maturity Onset Diabetes of the Young (MODY), as the specific phenotype of a non-ketotic, non-insulin-dependent diabetes of young onset, usually before 25 years of age, generally occurring in lean individuals, with an autosomal dominant transmission suggesting the molecular abnormality of a single gene. All these features have long been considered as diagnostic criteria for MODY. Abnormalities in many genes have been associated with MgD, among which three genes account for the most frequent genetic subtypes with a MODY phenotype, some are involved in neonatal diabetes or diabetes of infancy, and some are involved in syndromic diabetes (e.g., HNF1B syndrome, Wolfram syndrome, and mitochondrial diabetes). Abnormalities in other genes may be responsible for insulin resistance syndromes and syndromes of monogenic obesity, which may both be associated with diabetes, and for monogenic autoimmune diabetes [77]. Thus, beyond the "classical" MODY phenotype, MgD can be associated with multiple phenotypes.

Correlations between a patient's phenotype and the involved gene have been reported [78], but are questioned by the heterogenous phenotypes observed in patients harboring abnormalities of the same gene, and by the overlap of the diabetes phenotypes associated with different genes [79] (Figure 3). Moreover, next generation sequencing (NGS), which allows for the simultaneous screening of multiple genes and even the entire genome, leads to genetic diagnoses not predicted by the phenotypes [79–82]. Thus, from a clinical point of view, the main challenge for the clinician is to identify, among patients with diabetes, those who deserve genetic testing, i.e., to achieve the differential diagnosis with other diabetes subtypes.

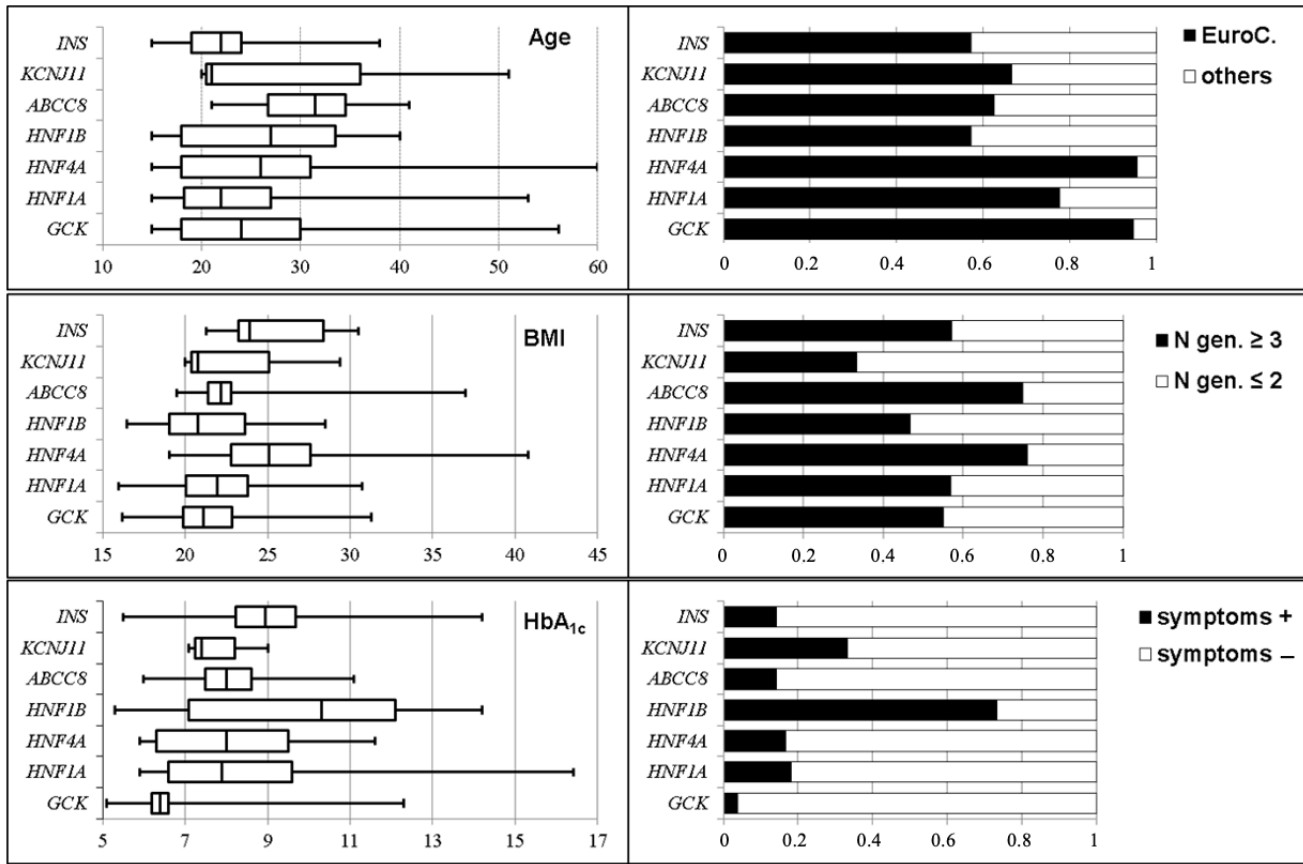

**Figure 3.** Distribution of the phenotypes according to genetic subtypes in patients with monogenic diabetes. Age (years), body mass index, BMI, (kg/m$^2$), HbA$_{1c}$ (%), ancestry (EuroC., Euro-Caucasian), numbers of affected generations, and presence of clinical symptoms. Except from symptoms and HbA$_{1c}$ in the patients with GCK-MODY, there are wide overlaps in all phenotypes among the genetic subtypes. Adapted from Donath et al. [79] with permission.

### 3.1. Epidemiology of MgD

In children and adults, population studies have shown that MgD accounts for at least 1–2% of all diabetes cases [80,83,84]. In most studies, molecular abnormalities of seven genes account for the majority of cases: glucokinase, *GCK*; transcription factors of the hepatocyte nuclear factor family, *HNF1A*, *HNF4A*, and *HNF1B*; *ABCC8* and *KCNJ11*, encoding the SUR-1 and Kir6.2 subunits of the ATP-dependent potassium channel of the pancreatic beta cell, respectively; and the m.3243A>G variant in the *MT-TL1* gene, which is responsible for Maternally Inherited Diabetes and Deafness (MIDD) syndrome.

### 3.2. Genotype/Phenotype Correlations of the Most Frequent MgD Subtypes
#### 3.2.1. GCK-MODY

GCK-MODY is the most frequent MgD, with an estimated prevalence of 0.3–1‰ in the general population [85,86]. The phenotype of GCK-MODY is very homogenous, which is unique among all diabetes subtypes. It is limited to a mild hyperglycemia (7 mmol/L on average) and a 6.5% average HbA$_{1c}$, present from birth, very stable in the long term, and detected in the proband and all the family members harboring the *GCK* pathogenic variant, i.e., a complete penetrance. However, the highly suggestive family history can be lacking, because the very mild hyperglycemia may have gone unrecognized. The prognosis of GCK-MODY is usually excellent, with no clinically significant complications of chronic hyperglycemia and no treatment is required [85]. However, up to 20% of patients with GCK-MODY are unnecessarily treated with oral hypoglycemic agents or even insulin, illustrating the frequent misdiagnosis of this MgD subtype [87]. Pregnancy is a privileged

situation to raise the diagnosis of GCK-MODY in women diagnosed with "gestational diabetes", although with no risk factors.

### 3.2.2. HNF1A-MODY

HNF1A-MODY is often diagnosed in adolescents or young adults, with a median age at onset of 20 years. Its penetrance increases with age, being nearly complete after 50 years. Clinical presentation at onset may suggest T1D in 25% of cases, but DKA is rare and the diagnosis can be made on the absence of T1D-associated antibodies and on the family history. In the majority of cases, the phenotype may suggest T2D, but usually with no features of metabolic syndrome [88]. A good sensitivity to sulfonylureas or glinides is maintained in most patients, allowing for weaning off insulin therapy [89]. The phenotype of HNF1A-MODY, particularly age at onset of diabetes, is highly variable, depending on the type and position of the *HNF1A* pathogenic variant [90], on genetic variants in *HNF1A* itself [91] and in other genes [92], the presence of body weight excess, and fetal exposition of the patient to maternal hyperglycemia [93,94].

### 3.2.3. HNF4A-MODY

HNF4A-MODY is a less frequent MgD subtype (5–10%). Its main characteristics are close to those of HNF1A-MODY, including a good sensitivity to sulfonylureas [95]. However, 50% of the neonates harboring an *HNF4A* variant have macrosomia, with a syndrome of hyperinsulinism and neonatal hypoglycemia in some. In the long term, hyperinsulinism vanishes and these children may develop diabetes. This sequential course is highly suggestive of HNF4A-MODY and can be observed in children who inherit the variant from their father, as well as their mother [96]. Thus, a personal or family history of neonatal severe hypoglycemia can help to raise this diagnosis.

### 3.2.4. HNF1B-Syndrome

HNF1B-syndrome is a rare MgD subtype (5% of cases), first described as the association of diabetes, suggesting a MODY phenotype, and renal cysts (RCAD phenotype) (reviewed in [97]). Beyond diabetes, this syndrome may include renal morphological abnormalities, renal functional defects that often progress to end-stage renal failure, abnormalities of the genital tract, morphological and functional abnormalities of the exocrine pancreas, abnormalities in liver tests, and an intellectual disability or autism spectrum disorder [98].

Over 50% of its cases are related to the microdeletion of chromosome 17 (17q12), which involves 15 genes, including *HNF1B* [99]. In 50% of cases, point pathogenic variants or the microdeletion occur de novo. Thus, an absence of family history should not exclude the diagnosis of HNF1B-syndrome.

The reported phenotypes of HNF1B-syndrome are highly variable, probably in part because of selection bias: for example, in children, the renal phenotype is in the foreground, with no overt diabetes. In contrast, among patients screened for MgD because of a phenotype suggesting classical MODY, cases of HNF1B-MODY have been diagnosed in patients with no renal phenotype [81,82].

Diabetes occurs in 50% of patients with an *HNF1B* molecular abnormality, at a mean age of 28 years. It can mimic T1D, with some cases being revealed by DKA, or T2D, with a more severe presentation in patients with the 17q12 deletion than in those with a point *HNF1B* variant [98].

### 3.2.5. Diabetes Associated with Pathogenic Variants of the Genes Encoding the K-ATP Channel Sub-Units

Pathogenic variants of *ABCC8* and *KCNJ11* have been associated with various phenotypes. Gain-of-function mutations in these genes may be responsible for severe neonatal diabetes, either permanent or transient, with a remission and a relapse during infancy or adolescence [100].

Patients with *ABCC8* pathogenic variants may present with a wide range of diabetes phenotypes, including mild abnormalities in glucose tolerance tests, gestational diabetes, late-onset diabetes suggesting T2D, or a severe presentation suggesting T1D [101,102].

Patients with diabetes related to the pathogenic variants of *ABCC8/KCNJ11*, even of a neonatal occurrence, are highly sensitive to sulfonylureas, even after insulin therapy of long duration [103,104].

### 3.2.6. Maternally Inherited Diabetes and Deafness Syndrome

The clinical presentation of this diabetes subtype is variable with an onset age between 20 and 40 years and a phenotype suggesting T1D in 20% of cases, but without diabetes-associated antibodies, or T2D in 80% of its cases, particular by the absence of metabolic syndrome and a BMI that is usually normal or even low [105]. Two main characteristics may help diagnosis: the disease is maternally transmitted, and diabetes may be associated with a wide array of extra-pancreatic manifestations, including neurosensory hearing loss, retinal macular dystrophy, myopathy, cardiomyopathy, glomerular nephropathy, peripheral neuropathy, and central nervous system defects [106]. However, the phenotype may be much more restricted because all these manifestations may be mild requiring a systematic evaluation to be detected, or even absent. Accordingly, the m.3243A>G variant of the MT-TL1 gene can be evidenced in patients referred for a clinical suspicion of MODY, in the absence of other overt manifestations [81,82].

### 3.3. Monogenic Diabetes Often Remain Misdiagnosed

In children [107,108] and adults [109], 50 to 90% of MgD cases are not diagnosed or misdiagnosed as T1D or T2D.

Several reasons may explain this situation, e.g., a lack of belief in the clinical benefits for the patients [110], a lack of access to genetic screening, a lack of knowledge about these rare diabetes subtypes, and the costs of genotyping, although studies have shown that it may be cost-effective [111]. One should note that the majority of MgDs are revealed in adults, with a phenotype that may suggest T2D, and these patients are usually followed by general practitioners who may not be well informed about MgD. A recent study showed that self-referral by the patients was the most efficient way to increase the diagnosis rate of MgD [112], underlying the potential for educating patients with diabetes on this topic.

While the extra-pancreatic features associated with diabetes may clinically suggest specific causes, the challenge of identifying patients who present with isolated diabetes, in whom genetic screening is warranted is probably one major pitfall that explains the frequent misdiagnoses of MgD. This raises two issues, i.e., the differential diagnosis with common diabetes subtypes and the strategy for genetic screening.

### 3.4. Differential Diagnosis with "Common" Diabetes Subtypes

In children and young individuals, mainly of Euro-Caucasian origin, algorithms based on the absence of beta cell autoantibodies at diabetes diagnosis and the persistency of insulin secretion (evaluated by plasma or urinary C-peptide concentration) at a distance from diabetes onset have been used to select patients who should be genotyped. In this context, the main differential diagnosis is T1D, and a MgD has been found in 4–15% of patients with no antibodies, or no antibodies and detectable insulin secretion [83,84,113,114]. Thus, it is recommended to exclude diagnoses of T1D in the absence of GADA, IA-2, and ZnT8 antibodies before genetic screening [115]. The long-term persistency of GADA and/or anti-IA-2 antibodies allows confirm a diagnosis of T1D in some patients, even long after the onset of diabetes, thus avoiding genetic screening [116,117].

In the recent years, polygenic scores have been developed to assess the risk of T1D. Models including T1D genetic risk scores have been proven efficient to identify children at a high risk for developing T1D [7,118] and to discriminate T1D from T2D in adult Euro-Caucasian patients [14,119]. These scores may also help to differentiate the fraction of patients with young-onset T1D but with no antibodies at onset, or those in whom antibodies

have disappeared at a distance from diagnosis, from those with MgD [120]. Moreover, they can help to identify patients in whom a diagnosis of MgD is likely but not confirmed by current techniques, who may benefit from extensive genotyping.

Differential diagnosis is more complex in patients with adult-onset diabetes, which is the most frequent situation. In a study, seven genes associated with MgD were screened using targeted NGS in 1564 patients, aged 15 years or more at diagnosis of diabetes, who were referred for a clinical suspicion of MgD, based on the absence of diabetes-associated antibodies, an age at diabetes diagnosis below 40 years, the absence of familial obesity, and the presence of a family history of diabetes [79]. MgD was identified in 16% of the patients, i.e., 15 times the expected prevalence in unselected adults with diabetes [80]. Using more stringent criteria would have led to a better specificity, but at the cost of a much lower sensitivity (Figure 4) [79].

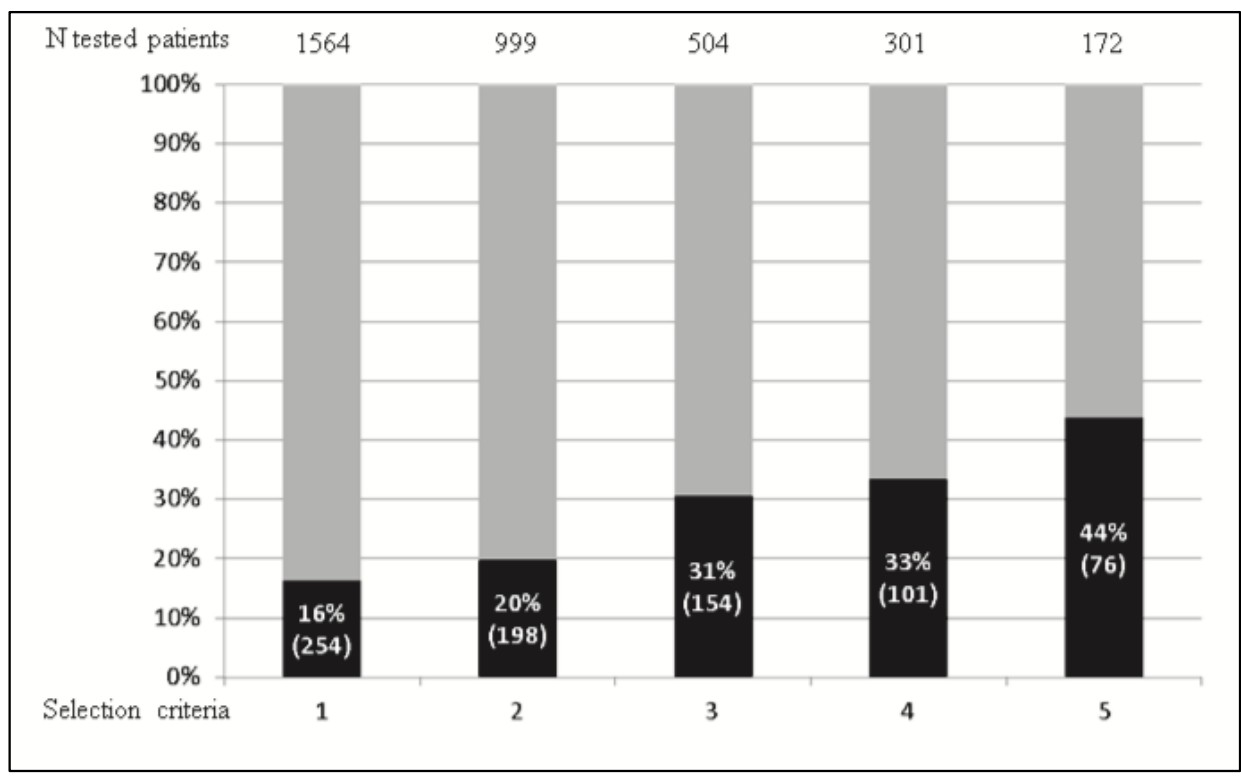

**Figure 4.** Diagnosis rate of monogenic diabetes according to the selection criteria used for genetic screening. Among 1564 patients referred for a clinical suspicion of monogenic diabetes, 254 (16%) cases (pathogenic/likely pathogenic variants) were confirmed. The total numbers of tested patients (upper part of the figure), frequency, and actual numbers of the patients with an identified monogenic diabetes (black boxes) are indicated according to the number of criteria used to select the patients to be genotyped, increasing from column 1 to 5. Using the most stringent criteria (column 5) would have led to reducing the number of patients to be tested by 90% and to increasing the diagnosis rate up to 44%, but would have led to missing 70% of the actual number of monogenic diabetes cases. Adapted from Donath et al. [40] with permission.

The great variability and overlaps between the phenotypes of MgD and common diabetes subtypes explain the difficulties of this differential diagnosis (Figure 5). For example, in the mentioned study, depending on the MgD subtype, 40–65% of the patients with an MgD were over 25 years of age at the onset of diabetes and 20% were overweight/obese [79]. In another study, among 4016 patients with adult-onset diabetes considered as T2D and no criteria suggesting MgD, a MgD was identified by systematic screening in 1.2%. Using the classical criteria for MgD screening, these cases would have been missed [80].

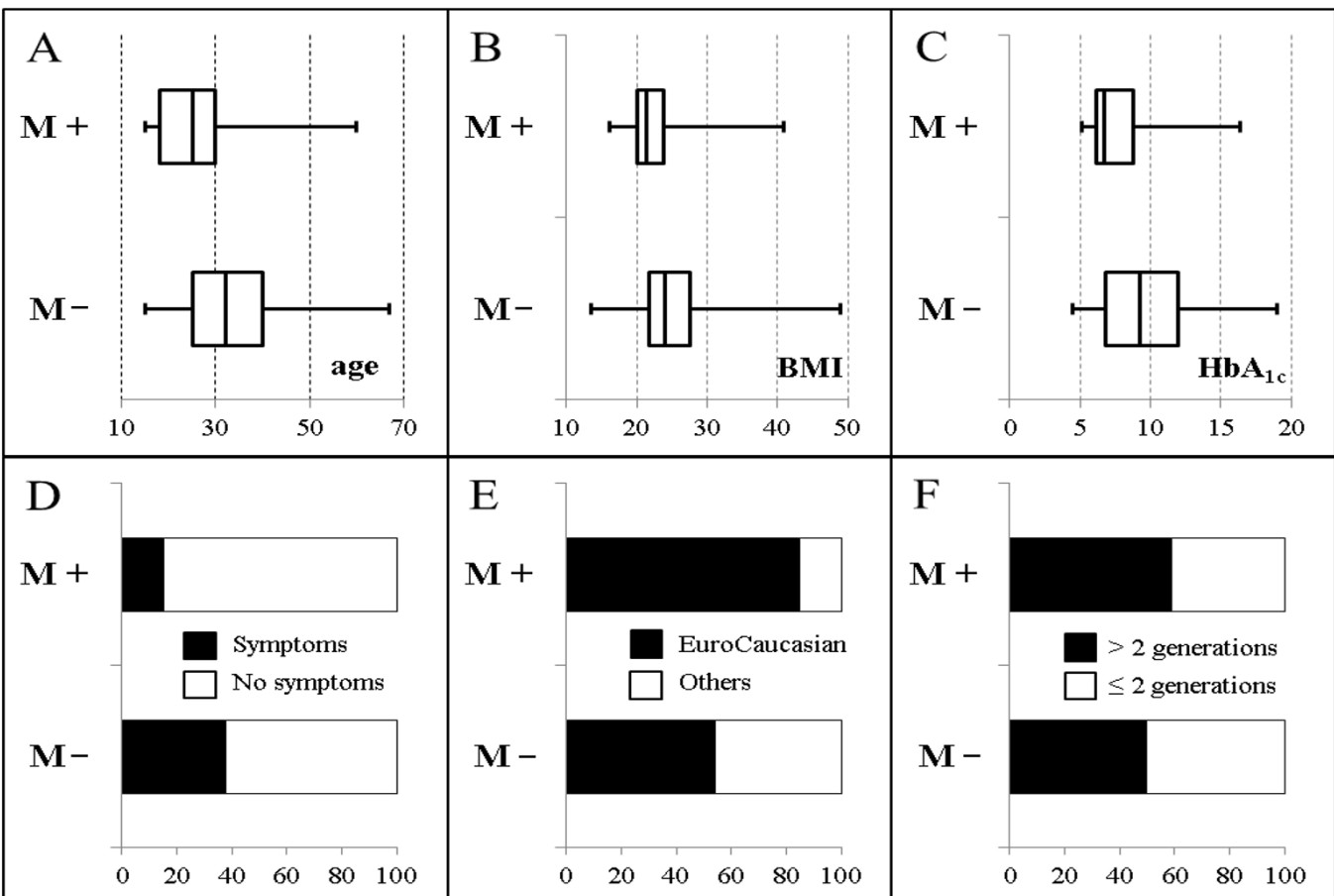

**Figure 5.** Main characteristics of antibody-negative patients with (M+, 249 patients) and without (M−, 1305 patients) monogenic diabetes, at the time of diabetes diagnosis. (**A**). Age (years); (**B**). Body mass index (kg/m$^2$); (**C**). HbA$_{1c}$ (%); (**D**). Presence of clinical symptoms of diabetes; (**E**). Ancestry (EuroCaucasian *vs* non EuroCaucasian); (**F**). Number of generations with diabetes. Adapted from Donath et al. [40] with permission.

Similarly, a large study reported that the pathogenic/likely pathogenic variants (P/LP) of five actionable MODY genes (*GCK*, *HNF4A*, *KCNJ11*, *HNF1B*, and *ABCC8*, but surprisingly not *HNF1A*) were significantly more frequent in patients with adult-onset T2D (2%) than in non-diabetic controls (1%) [121]. At diagnosis of T2D, those with P/LP variants were younger and leaner than those with no variants, but no case occurred before the age of 25 years and there were no differences in the family history of T2D between the two groups. Therefore, these cases would not have been identified by the classical MODY criteria. Moreover, the patients with P/LP variants, including those with *GCK* variants, were more frequently treated with insulin and less with metformin, suggesting that, in some cases, the treatment was spuriously orientated by the phenotype of the patients, not by the etiology of the diabetes [121].

### 3.5. Which Strategy for the Genetic Diagnosis of MgD?

Next generation sequencing techniques are now available in routine practice and allow for the screening of multiple genes (targeted NGS), or even the whole exome/genome.

One approach, mostly used in practice, consists of selecting patients with "atypical" phenotypes for genetic testing. For example, in the Rare and Atypical Diabetes Network study (ClinTrials.gov, NCT05544266), whole genome sequencing will be performed in patients with diabetes harboring one or several phenotypes, such as T2D diabetes of early onset; T2D, gestational diabetes, lipodystrophy, insulin resistance or polycystic ovary

syndrome occurring in lean individuals; Mendelian inheritance; syndromic diabetes; KPD. Those with no identified pathogenic variant will undergo extensive clinical and biological phenotyping, and cluster analyses will be performed to define "atypical diabetes" subtypes [122].

As mentioned, NGS techniques may identify MgD cases not predicted by the phenotype of the patients and may identify pathogenic variants in the patients with "common" phenotypes. Since an accurate diagnosis conveys benefits for patients, some authors have suggested that whole genome sequencing could be performed in all patients with diabetes. Such genotyping could generate additional data, e.g., a predisposition to diabetes complications or other diseases and pharmacogenetic information [123].

Yet, these approaches are limited in routine practice by the difficult and time-consuming processes of confirming the pathogenicity of an identified variant [124]. According to current recommendations [125], this crucial step in diagnosis is based on a series of analyses, such as the type of the variant, the prevalence of the variant in the general population, the use of several prediction algorithms, co-segregation studies of the variant and the phenotype in families, analysis of previous reports of the variant, and bioassays.

### 4. Conclusions

Many epidemiological, clinical and pathophysiological data show that the "typical" forms of diabetes (namely T1D, T2D, and monogenic diabetes) are historical and too restrictive. Clinical and biological traits reveal a large heterogeneity in the phenotypes of each of these diabetes subgroups and large overlaps in their phenotypes, which make the etiological diagnosis of diabetes difficult. In recent years, diabetes subtypes have been identified within each diabetes subgroup, which may be associated with distinct genetic and pathophysiological backgrounds, and with different rates of progression, risks of complications, and responses to treatment, leading to the application of the endotype concept to diabetes. The clinical consequences of these sub-classifications in routine practice, especially the opportunity to obtain precision medicine in diabetes, remains, however, to be determined. Yet, these observations should lead to a systematic approach to the etiological diagnosis of diabetes, including careful clinical and biological phenotyping at onset of diabetes.

**Author Contributions:** D.D.-L. and J.T. conceived the work, wrote the draft of the manuscript and revised it, and approved its final version. All authors have read and agreed to the published version of the manuscript.

**Funding:** This research received no external funding.

**Institutional Review Board Statement:** Not applicable.

**Informed Consent Statement:** Not applicable.

**Data Availability Statement:** Not applicable.

**Acknowledgments:** Part of this work was presented at the 39èmes Journées Nicolas Guéritée d'Endocrinologie et Maladies Métaboliques, 2019, Paris, France, and published in Mises au point Cliniques d'Endocrinologie, Nutrition et Métabolisme, P. Chanson, J. Orgiazzi and J.L. Thomas, Eds., Les Editions de Médecine Pratique, Tulle, France.

**Conflicts of Interest:** The authors declare no conflict of interest.

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
