# Peer review of "The Etiological Diagnosis of Diabetes: Still a Challenge for the Clinician"

_endocrines, doi:10.3390/endocrines4020033_

Round 1
Reviewer 1 Report
In this manuscript, the authors discuss the challenge of diagnosing distinct subtypes of diabetes based on their clinical presentations. The authors elaborate on this under specific subtypes of diabetes and propose that a systematic evaluation of multiple phenotypes would resolve this problem. Overall, this provides an excellent overview of the classification of diabetes and should be of great interest to researchers and the general public interested in metabolic diseases.
I only have some minor suggestions to be considered by the authors:
Line 36, how about replacing "many" with "important"
Line 37, "in many situations" seems unneeded
Line 43, perhaps delete "although....."
Line 408, "of age" missing
Line 577, perhaps delete "although..."
Line 689, I don't think I understand the meaning of the word "caricatural" in this context
Author Response
Line 36, how about replacing "many" with "important" : this has been done
Line 37, "in many situations" seems unneeded : It has been deleted
Line 43, perhaps delete "although....." : this has been done
Line 408, "of age" missing : this has been corrected
Line 577, perhaps delete "although..." : this has been done
Line 689, I don't think I understand the meaning of the word "caricatural" in this context : we have replaced the term caricatural by « too restrictive » for a better understanding
Reviewer 2 Report
The review comprehensively examines current literature on the challenges associated with the diagnosis and classification of diabetes, and thus holds immense clinical significance.
- To enhance the clarity and organization of the data presented, I recommend that the authors incorporate Tables and Figures.
- The section on MODY should be concise, focusing solely on the latest advancements in our understanding of the different types of MODY listed in section 2.3.
- Lines 491-494 are incomprehensible and require clarification.
- In the discussion section, I suggest including information on clinical trials (CT) exploring the future of "Atypical Diabetes," such as the RADIANT trial, which aims to discover novel mechanisms and biomarkers for this disease.
- It is advisable to avoid using terminologies such as "so-called" and "actually" in the review.
Author Response
- To enhance the clarity and organization of the data presented, I recommend that the authors incorporate Tables and Figures : we have incorporated one figure suggesting algorithms to etiological diagnosis of diabetes, based on clinical presentation at onset (figure 1) to offer a more comprehensive view to the readers.
- The section on MODY should be concise, focusing solely on the latest advancements in our understanding of the different types of MODY listed in section 2.3. : we have cut the section reporting the clinical presentation of the different MODY subtypes. The description of the phenotypic traits that characterize each subtype and may lead to the diagnosis have been maintened.
- Lines 491-494 are incomprehensible and require clarification : these two sentences have been removed.
- In the discussion section, I suggest including information on clinical trials (CT) exploring the future of "Atypical Diabetes," such as the RADIANT trial, which aims to discover novel mechanisms and biomarkers for this disease : we have incorporated a small paragraph describing the RADIANT trial and added one reference (ref 125)
- It is advisable to avoid using terminologies such as "so-called" and "actually" in the review : these terms have been deleted